# Tandem-repeat protein domains across the tree of life

Kristin K. Jernigan[1] and Seth R. Bordenstein[2,3]

[1] Department of Cell and Developmental Biology, Vanderbilt University, Nashville, TN, USA
[2] Department of Biological Sciences, Vanderbilt University, Nashville, TN, USA
[3] Department of Pathology, Microbiology, and Immunology, Vanderbilt University, Nashville, TN, USA

## ABSTRACT

Tandem-repeat protein domains, composed of repeated units of conserved stretches of 20–40 amino acids, are required for a wide array of biological functions. Despite their diverse and fundamental functions, there has been no comprehensive assessment of their taxonomic distribution, incidence, and associations with organismal lifestyle and phylogeny. In this study, we assess for the first time the abundance of armadillo (ARM) and tetratricopeptide (TPR) repeat domains across all three domains in the tree of life and compare the results to our previous analysis on ankyrin (ANK) repeat domains in this journal. All eukaryotes and a majority of the bacterial and archaeal genomes analyzed have a minimum of one TPR and ARM repeat. In eukaryotes, the fraction of ARM-containing proteins is approximately double that of TPR and ANK-containing proteins, whereas bacteria and archaea are enriched in TPR-containing proteins relative to ARM- and ANK-containing proteins. We show in bacteria that phylogenetic history, rather than lifestyle or pathogenicity, is a predictor of TPR repeat domain abundance, while neither phylogenetic history nor lifestyle predicts ARM repeat domain abundance. Surprisingly, pathogenic bacteria were not enriched in TPR-containing proteins, which have been associated within virulence factors in certain species. Taken together, this comparative analysis provides a newly appreciated view of the prevalence and diversity of multiple types of tandem-repeat protein domains across the tree of life. A central finding of this analysis is that tandem repeat domain-containing proteins are prevalent not just in eukaryotes, but also in bacterial and archaeal species.

## INTRODUCTION

While many functional protein domains exist, tandem-repeat domains are one of the most abundant classes of protein–protein interaction domains (*Heringa, 1998*; *Marcotte et al., 1999*). Tandem-repeat domains are comprised of 'tandem' arrays of repeating units of approximately 20–40 amino acids that contain simple structural motifs, such as $\alpha$-helices or $\beta$-sheets (*Kobe & Kajava, 2000*). These domains can be classified as having either an open structure with a variable number of repeats or a closed structure with a fixed number of repeats. The repeat domains that fall in the former category are typically composed of

Corresponding author
Seth R. Bordenstein,
s.bordenstein@vanderbilt.edu

3–20 or more repeats, and the resulting structural formation determined by the tandem array of these protein motifs provides a platform for protein–protein interactions (*Grove, Cortajarena & Regan, 2008*). Some of the most common open-structure repeat domains include the ankyrin (ANK), tetratricopeptide (TPR), and armadillo (ARM) domains.

Tandem-repeat containing proteins are present in all domains of life and function in nearly every cellular process from transcriptional regulation in the nucleus to cell adhesion at the plasma membrane (*Andrade et al., 2001*). Based on our recent study on the distribution of ANK-containing proteins across the tree of Life, ANK domains are more common in bacteria (51% of strains) and archaea (11% of strains) than previously recognized (*Jernigan & Bordenstein, 2014*). Here, we set out to determine the distribution of TPR and ARM repeats across the three domains of life to examine three general patterns: (i) the origins and distribution of these repeats across the tree of life, (ii) the correlated presence of these repeats in each proteome, and (iii) the strength of associations between the presence of these protein domains and the taxonomic lifestyle or phylogenetic relationships of the organisms that they inhabit. We establish that these tandem-repeat proteins are not only present, but abundant in all domains of the universal tree of life.

The ANK repeat is a 33 amino acid motif that originally was discovered in *Saccharomyces cerevisiae*, *Schizosaccharomyces pombe*, and *Drosophila melanogaster* and was named after a human protein by the same name (*Breeden & Nasmyth, 1987*; *Mosavi et al., 2004*). The structure of a single motif begins with a $\beta$-turn that precedes two antiparallel $\alpha$-helices and ends with a loop that feeds into the next repeat (Fig. 1A). These motifs stack one upon another to form an ANK domain and provide an interface for interacting with other proteins (*Gorina & Pavletich, 1996*; *Sedgwick & Smerdon, 1999*). ANK-containing proteins have been identified in all domains of life, and special attention has recently been paid to the role of ANK-containing proteins in host-microbe interactions in bacterial species such as *Legionella pneumophila* (*Al-Khodor et al., 2010*; *de Felipe et al., 2008*), *Anaplasma phagocytophilum* (*JW, Carlson & Kennedy, 2007*), and *Ehrlichia chaffeenis* (*Zhu et al., 2009*).

The TPR repeat is 34 amino acids long and is composed of two $\alpha$-helices producing an $\alpha$-helix-turn-$\alpha$-helix motif (Fig. 1A) (*Das, Cohen & Barford, 1998*). First identified in yeast cell cycle proteins, it was coined the tetratricopeptide repeat for its 34 amino acid sequence (*Hirano et al., 1990*; *Sikorski et al., 1990*). A typical TPR domain contains between 3 and 16 TPR repeats and ends with one additional resolving $\alpha$-helix that is thought to provide stability to the protein domain (*D'Andrea & Regan, 2003*). TPR-containing proteins occur in all domains of life (*Cerveny et al., 2013*; *Ponting et al., 1999*), but no systematic investigation has specified their general distribution and incidence across the universal tree.

ARM repeats, at 42 amino acids, are composed of three $\alpha$-helices (Fig. 1A) (*Tewari et al., 2010*). This domain was first identified in the *Drosophila melanogaster* segment polarity protein, Armadillo (*Peifer, Berg & Reynolds, 1994*). In our analysis, other repeat domains with similar sequence, structure, and function are classified under the ARM repeat superfamily, including the HEAT repeat (*Andrade et al., 2001*; *SUPERFAMILY*). The HEAT repeat, named after the first proteins identified to contain this repeat (i.e., Huntingtin,

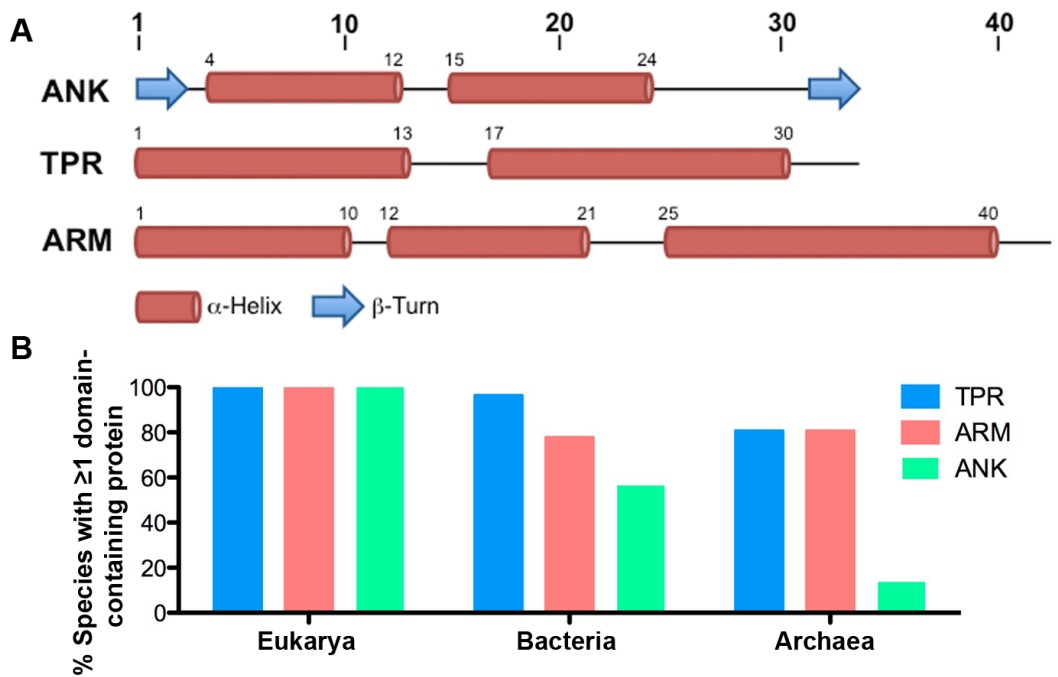

**Figure 1** **ANK, TPR, and ARM repeat structure and incidence across all domains of life.** (A) Schematics of the structures of the ANK (*Mosavi et al., 2004*), TPR (*Main et al., 2005*), and ARM (*Parmeggiani et al., 2008*) repeats. (B) Table indicating the percent of species in each taxonomic domain that contain ≥1 TPR, ARM, or ANK-containing protein.

Elongation factor 3, regulatory subunit A of Protein Phosphatase 2A, and Target of rapamycin) is composed of two $\alpha$-helices (*Andrade & Bork, 1995*; *Andrade et al., 2001*). Although composed of a different number of $\alpha$-helices, both repeats produce a similar concave surface important for protein interactions. Sequence analysis indicates that both repeats contain seven conserved amino acids (*Andrade et al., 2001*; *Cingolani et al., 1999*; *Eklof Spink, Fridman & Weis, 2001*; *Lee et al., 2003*).

All three repeat domains are composed of multiple repeated units of relatively simple protein motifs that impart important cellular functions. To the best of our knowledge, this is the first comprehensive analysis of multiple repeat domains across the tree of life that additionally shows how their abundance associates with phylogenetic history and lifestyle.

## MATERIALS AND METHODS

### ANK, TPR and ARM-containing protein data acquisition and analysis

All genome information was obtained from the SUPERFAMILY v1.75 database (*SU-PERFAMILY*; *Wilson et al., 2009*), including the taxonomy, and number of ANK, TPR and ARM-containing proteins. At the time of the analysis, the SUPERFAMILY database contained protein domain information on 2,489 strains, where there can be more than one strain representing a single phylogenetic species. This database is an archive of structural and functional domains in proteins of sequenced genomes (*Wilson et al.,*

*2009*), which are annotated using hidden Markov models through the SCOP (Structural Classification of Proteins) SUPERFAMILY protein domain classification (*Gough et al., 2001*; *SUPERFAMILY*). We note appropriate caution that ANK, TPR, and ARM domains are called based on a computational framework and are not experimentally confirmed. NCBI's Genome Resource was used to obtain total gene and protein numbers for each organism in the analysis (Table S1). To determine the percent of an organism's total protein number (proteome) that is composed of ANK/TPR/ARM-containing proteins, the number of ANK/TPR/ARM-containing proteins was divided by the total number of proteins and multiplied by 100. Only organisms with available total protein information were used in this analysis. For these analyses, an average of the number and/or percent of ANK/TPR/ARM-containing proteins for all strains of the same species were used.

## 16S rRNA phylogenetic tree and independence analysis

16S rRNA sequences from one randomly selected species in each class of Fig. 3 were aligned by MUSCLE in Geneious Pro 5.6.2. Prior to building the tree, a DNA substitution model for the alignment was selected using jModelTest version 2.1.3 (*Darriba et al., 2012*; *Guindon & Gascuel, 2003*). A Bayesian phylogenetic tree was generated by MrBayes using the HKY85 model of DNA sequence evolution (*Hasegawa, Kishino & Yano, 1985*; *Huelsenbeck & Ronquist, 2001*; *Ronquist & Huelsenbeck, 2003*). For testing phylogenetic independence of ANK/ARM/TRP-containing proteins in bacteria, the PDAP program in Mesquite was used to generate independent contrasts for the data in Fig. 5 (*Maddison & Maddison, 2006*; *Midford, Garland & Maddison, 2005*). Phylogenetic Independence version 2.0 (*Reeve & Abouheif, 2003*) performed the Test For Serial Independence (TFSI) based on the Bayesian tree.

## RESULTS

### The incidence of tandem-repeat containing proteins decreases from Eukaryotes > Bacteria > Archaea

Of the 2,489 proteomes analyzed, 1,911 are from the domain Bacteria, 445 are from the domain Eukarya, and 133 are from the domain Archaea. For many species of bacteria and archaea, more than one proteome per species is present within the SUPERFAMILY database. We report that all eukaryotic species contain at least one TPR, ANK, and ARM-containing protein (except *Saccharomyces cerevisiae CLIB382*, which lacks an ANK-containing protein and a completely annotated genome) (Fig. 1B, Table S1). Nearly all species of bacteria (96.42%, 970/1,007) and a majority of the archaeal species (80.73%, 88/109) contain at least one TPR-containing protein. Similarly, we found that a majority of bacterial and archaeal species (77.73% and 80.73%, or 782/1,007 and 88/109, respectively) contain at least one ARM-containing protein. ANK-containing proteins were the least abundant protein domain with 56.6% (569/1,007) of bacterial species and only 13.76% (15/1,007) of archaeal species containing one or more ANK-containing proteins (Fig. 1B).

When we group organisms into genera to account for biases in species selected for sequencing, we find no major changes in the distribution and incidence of these proteins.

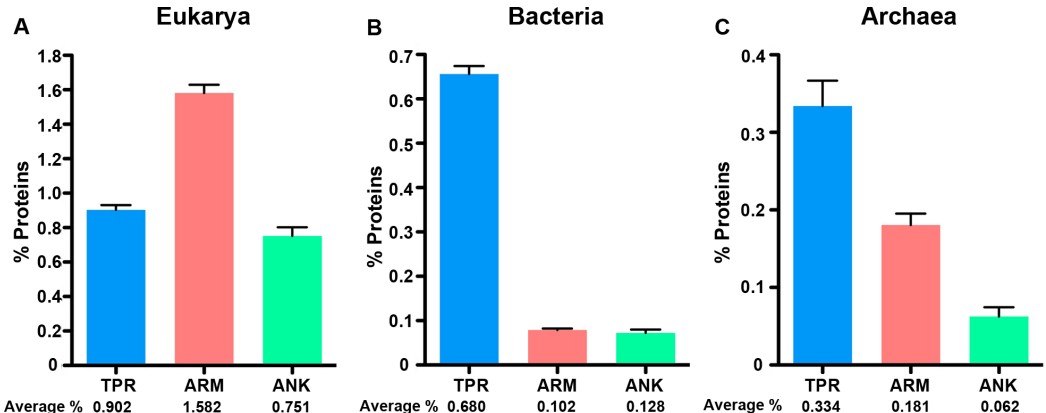

**Figure 2 ANK, TPR, and ARM-containing proteins analysis across all domains of life.** Bar graphs of the average percent of the species in the domains. An average percent of ANK, TPR, or ARM-containing proteins for all organisms of the same species was used for these analyses. (A) Eukarya, (B) Bacteria, and (C) Archaea that have one or more ANK, TPR, and ARM-containing proteins. The average percent of protein domain composition is listed below the graph. (A) Mann Whitney U-test $p < 0.0001$ for all comparisons, ANOVA $p = 5.79E-40$. (B) Mann Whitney U-test $p < 0.0001$ for all comparisons, ANOVA $p = 2.3E-203$. (C) Mann Whitney U-test, $p < 0.005$ for all comparisons, ANOVA $p = 9.85E-07$. Error bars represent standard error.

98.2% of bacterial genera (531/541) and 77.9% of archaeal genera (53/68) contain species that encode at least one TPR-containing protein. 76.8% of bacterial genera (417/541) and 77.1% of archaeal genera (54/68) contain species that encode at least one ARM-containing protein. Finally, 56.7% of bacterial genera (307/541) and 8.8% of archaeal genera (6/68) contain species that encode at least one ANK-containing protein.

Next, we normalized the percent of the proteome dedicated to each protein type across the three cellular domains to compare differences in protein abundance (Table S1, Fig. 2). After normalizing for proteome size, we find that eukaryotes consistently have higher amounts of each protein domain than the other two domains, but the relative fraction of each protein in the specific domains varies. For instance, the fraction of ARM-containing proteins is approximately double the abundance of TPR and ANK-containing proteins in eukaryotes (Fig. 2A, Mann–Whitney U, $p < 0.0001$, ANOVA, $p = 5.79E-40$); but the enrichment patterns switch in bacteria and archaea, in which the fraction of TPR-containing proteins are enriched relative to ARM and ANK-containing proteins (Bacteria: Mann Whitney U-test $p < 0.0001$ for all comparisons, ANOVA $p = 2.3E-203$. Archaea: Mann Whitney U-test, $p < 0.005$ for all comparisons, ANOVA $p = 9.85E-07$).

## The variation in intraproteomic abundance of each repeat domain is correlated in bacteria

The majority of bacterial taxa have ANK, TPR, and ARM domains (Fig. 1), which permits an analysis of whether their relative and absolute abundances within each proteome are correlated positively or negatively. We performed non-parametric correlation analyses on the normalized percent and absolute number of each protein per species. Results specify significant correlations for (i) the intraproteomic relative abundance of

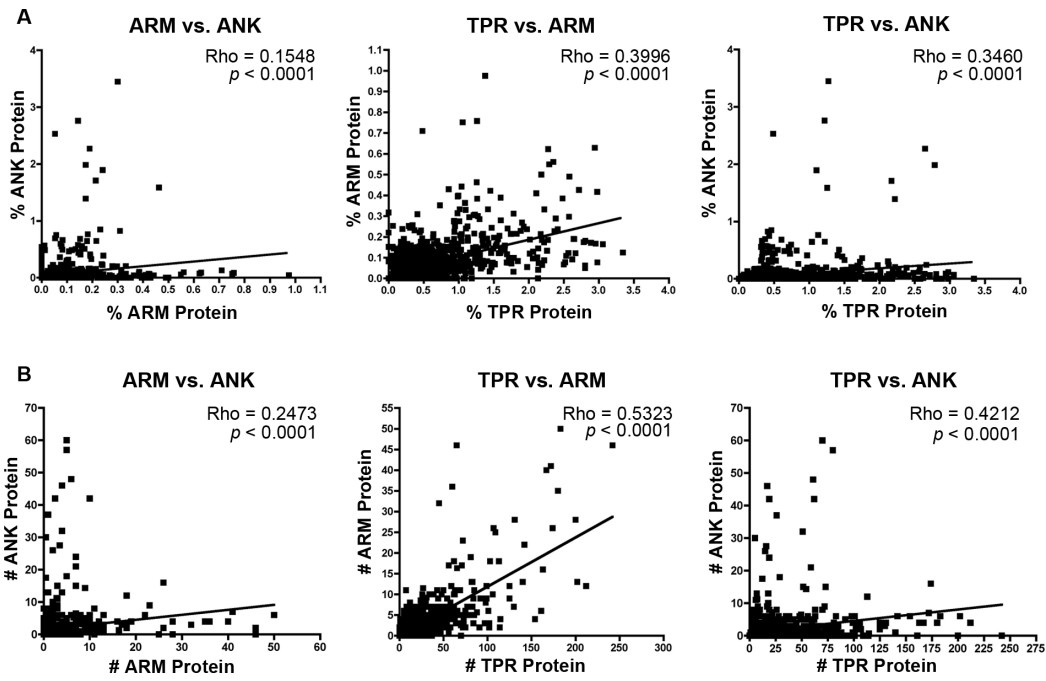

**Figure 3 Correlation analysis.** (A) Percent protein composition of TPR, ARM, or ANK-containing proteins in bacterial species and (B) Number of TPR, ARM, or ANK-containing proteins in bacterial species.

each domain-contain protein across numerous bacterial species (Fig. 3A) and (ii) the intraproteomic absolute abundance (i.e., total number of domain-containing proteins) (Fig. 3B). The stronger correlations, as measured by the Rho values, occur in analyses when the TPR domain is compared to the ANK and ARM domains.

To determine which bacterial taxa are enriched for ANK, TPR, and ARM-containing proteins, we analyzed 24 bacterial classes composed of 953 proteomes for the percent of species in each class that have one or more domain containing proteins (Fig. 4). As noted above, most bacterial species have at least one TPR domain, and all of the classes analyzed contain species with one ARM domain (Fig. 4). While the percent of species in a bacterial class that have one or more domains can vary widely, the Mollicutes curiously show low abundance for each of the domains (range of 3.1%–12.5%, Fig. 4). We discuss this outlying taxa below.

The bacteria that harbor an enriched fraction of ANK, TPR, and ARM-containing proteins, as defined by the species that fall within the top 15% of protein domain abundance, are shown adjacent to the bacterial tree in Fig. 5. For the ANK and ARM-containing proteins, the percent of the species that meet this enrichment cutoff are not significantly associated with the bacterial tree such that the normalized abundance of ANK and ARM-containing proteins per proteome across the 24 classes is independent of phylogenetic relatedness (ANK $p = 0.5590$, ARM $p = 0.3770$, PI test, (*Reeve & Abouheif, 2003*)) (Fig. 5). In contrast, normalized TPR-containing protein abundance is phylogenetically dependent ($p = 0.0270$, PI test). This association appears to be due to the abundance

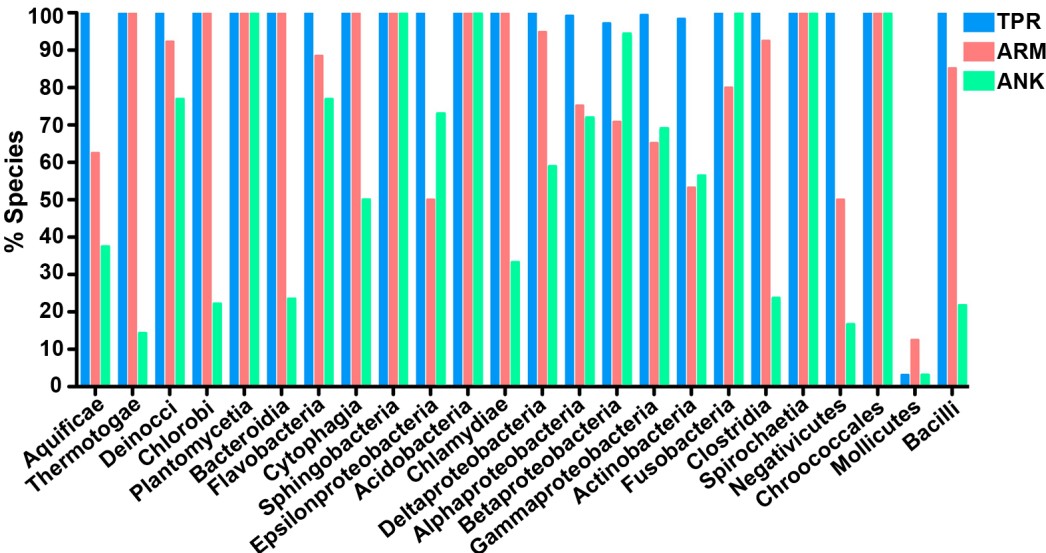

**Figure 4 TPR, ANK, and ARM-containing proteins analysis across bacterial classes.** Bar graph of the percent of species in bacterial classes that contain ≥1 TPR, ARM, or ANK-containing proteins (only classes with 5 or more represented species were included in this analysis). Order of classes reflect phylogenetic relationships depicted in Fig. 5A.

of TPR-containing proteins in the classes Bacteroidia, Flavobacteria, Cytophagia, and Sphingobacteria, which are all part of the phylum Bacteroidetes. To test if the phylogenetic dependence of the TPR abundance is robust to small changes in the cutoff value, we repeated the analysis with the top 10% and 20% of bacterial taxa that harbor an enriched fraction of TPR-containing proteins. We find phylogenetic dependence at the 10% cutoff ($p = 0.002$, PI test), but not at the 20% cutoff ($p = 0.0780$, PI test) (data not shown).

Finally, the percent of species in each class with an abundance of ANK and ARM-containing proteins is independent of phylogenetic history (Fig. 5). These values are correlated to the percent of species with a high abundance of TPR-containing proteins, but are not correlated to each other (Fig. 5C).

## ARM and TPR domains are not enriched in host-associated bacteria

We reported in an article published previously in this journal that host-associated bacteria (including obligate intracellular bacteria that replicate strictly within host cells and facultative host-associated bacteria) are enriched in ANK-containing proteins in comparison to free living bacterial species that solely replicate outside of host cells (*Jernigan & Bordenstein, 2014*). To test if host-associated bacterial species are preferentially enriched in ARM-containing proteins, we examined the abundance of ARM-containing proteins per strain. The percent of bacterial species with a high ARM-containing protein composition rapidly declines as the percent of ARM-containing proteins increases (Fig. S1). Thus, we assigned one of three lifestyle annotations (free-living (FL), facultative host-associated (FHA) and obligate intracellular (O) bacteria) for all bacterial species with at least 0.2% of their proteome dedicated to ARM-containing proteins (representing the

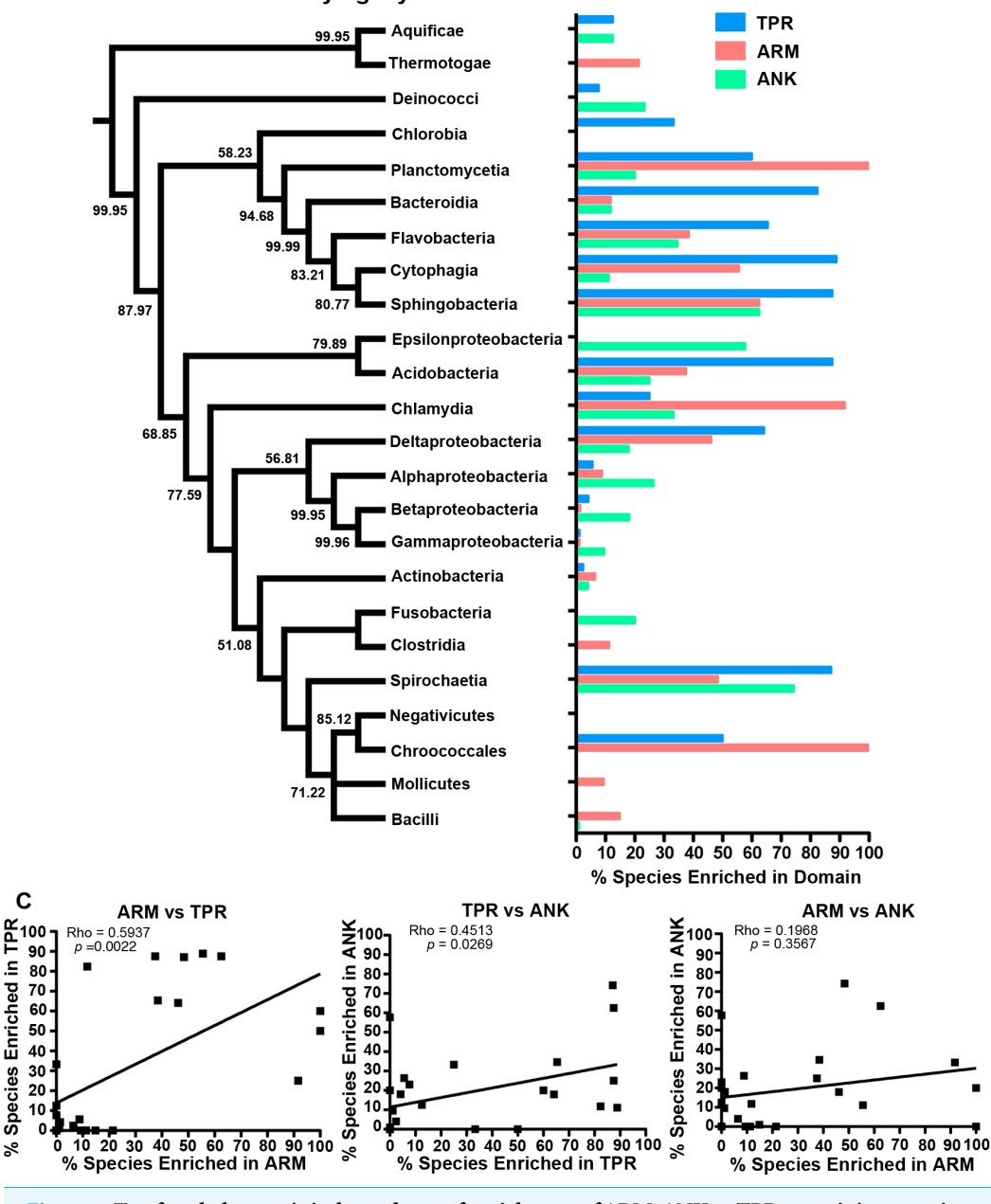

**Figure 5** **Test for phylogenetic independence of enrichment of ARM, ANK or TPR-containing proteins in bacteria.** (A) Consensus phylogeny of 16S rRNA sequences from one species (randomly selected) in each class. (B) The percent of species per class that fall within the top 15% most enriched species in the dataset. Only classes with 5 or more represented species were included in this analysis. (C) Spearman correlation analysis of the percent of species within bacterial classes enriched in each domain.

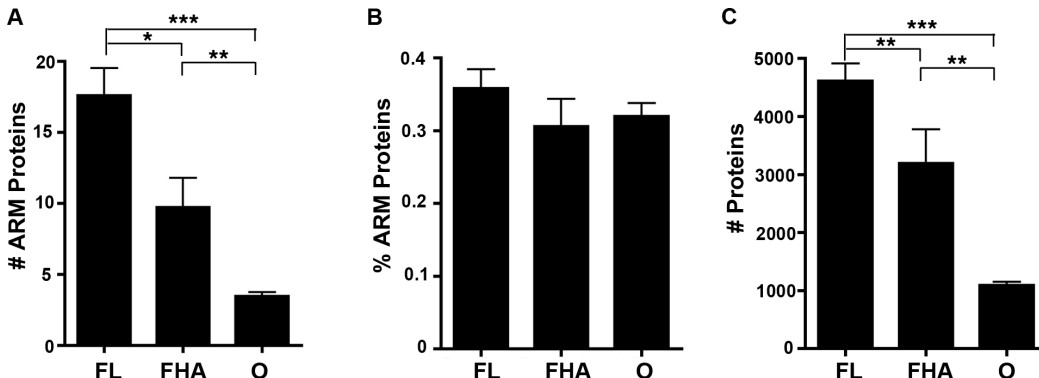

**Figure 6 Lifestyle analysis of bacterial species enriched for ARM-containing proteins.** (A-C) Error bars represent standard error and *$P < 0.05$, **$P < 0.01$, **$P < 0.0001$. (A) Bar graph of the average number of ARM-containing proteins in species of free-living (FL), facultative host-associated (FHA) and obligate intracellular (O) bacteria for those species with at least 0.2% of their proteome composed of ARM-containing proteins. (2-tailed Mann-Whitney U; ANOVA, $P = 0.00005$). (B) Bar graph of the average percent of the proteome composed of ARM-containing proteins in species of FL, FHA and O bacteria with at least 0.2% of their proteome composed of ARM-containing proteins. (All comparisons, $P > 0.05$, 2-tailed Mann-Whitney U; ANOVA, $P = 0.446$). (C) Bar graph of the average number of proteins in the proteomes of species of FL, FHA and O bacteria with at least 0.2% of their proteome composed of ARM-containing proteins (2-tailed Mann-Whitney U; ANOVA, $P = 1.98E-10$).

top 7.5% of species enriched in ARM-containing proteins) (Table S2). We selected this cut off for annotation of the lifestyles to analyze bacterial species whose proteome is enriched in ARM-containing proteins. When analyzing absolute protein abundance, we found that host-associated microbes (i.e., obligate intracellular and facultative host-associated) contain, on average, a lower number of ARM-containing proteins than those that are free-living (Fig. 6A). However, once normalized for genome size, the difference is not statistically significant (Fig. 6B).

Similarly, to determine if host-associated bacteria are enriched in TPR-containing proteins, we assigned these three lifestyle annotations for all bacterial species with at least 1.5% of their proteome dedicated to TPR-containing proteins (representing the top 8.2% of species enriched in TPR-containing proteins) (Table S3). The percent of bacterial species also declines as the cutoff percentage of TPR-containing proteins nears 1.5% (Fig. S2), so we analyzed the lifestyle of species within this cutoff. Once again, the proteomes of host-associated bacteria are not enriched in TPR-containing proteins (Fig. S3).

## TPR domains are not enriched in pathogenic bacteria

Roles for bacterial TPR-containing proteins in virulence mechanisms have gained increasing attention over the past decade (*Cerveny et al., 2013*), spanning adhesion of *Francisella tularensis* to host cells through pili formation (*Chakraborty et al., 2008*), translocation of *Pseudomonas aeruginosa* effectors into host cells (*Broms et al., 2006*), and inhibition of phagolysosomal degradation of *Mycobacterium tuberculosis* (*Chao et al., 2010*). Thus, we hypothesized that the relative abundance of TPR-containing proteins in bacterial proteomes would be associated with increased virulence. To determine if

pathogenic bacteria were enriched in TPR-containing proteins, we generated a list of bacterial species with TPR-containing protein enriched proteomes (i.e., species with a TPR proteome composition greater than 2.0%, $N = 48$ species, in comparison to the bacteria-wide average of 0.66%). Based on primary literature searches, we categorized these taxa by their lifestyle and whether or not they cause human disease. Interestingly, only 22.9% of these taxa enriched in TPR-containing proteins were pathogenic (Table S4), and those species that are pathogenic have not been documented to harbor virulence factors that contain TPR domains.

We also analyzed the number of TPR-containing proteins present in bacterial species known to harbor TPR containing-virulence factors as described by *Cerveny et al. (2013)* (Table S5). We report that these species have a relatively reduced number and proteome composition of TPR-containing proteins (Table S5). Many of these species contained fewer TPR-containing proteins than the average of all species in the data set (average $13.2 \pm 8.16$ SD vs. $23.8 \pm 28.23$ SD for the whole data set, $T$-test, $p = 0.0008$).

## DISCUSSION

A central finding from these results is that tandem repeat domain-containing proteins are prevalent not just in eukaryotes, but also in bacterial and archaeal species. A great majority of bacterial and archaeal taxa analyzed (96.42% and 80.73%, respectively) contain at least one TPR-containing protein. In bacterial species, 85.6% of all species analyzed contain at least two repeat domains, and of those species, over half (52%) contain all three repeat domains. Interestingly, Mollicutes are the only bacterial class with a reduced protein composition for all three domain-containing proteins. Of the 32 Mollicutes species, one of these species contains an ARM-containing protein, one contains an ANK- containing protein, and four contain a TPR-containing domain. This observation may, in part, be due to the reduced genome sizes and simplified metabolic pathways of Mollicutes, though many obligate intracellular bacteria with similar traits have these protein domains. Mollicutes are relatively unique among bacteria in that they also lack peptidoglycan, and instead have cholesterol-containing cell membranes (*Krieg et al., 2011*). Although it is uncertain if the highly reduced abundance of tandem repeat domain-containing proteins in this group is linked to the absence of peptidoglycan, there is evidence for an association in *Escherichia coli* in which the TPR-containing lipoprotein LpoA stimulates the major peptidoglycan synthase PBP1A (*Jean et al., 2014*).

We express appropriate caution that since certain strains of bacteria with relevance to human health have received extra attention in sampling, it is possible that the SUPERFAMILY dataset used in the analyses is not representative of the microbial diversity of the natural world, but rather is enriched in bacterial species that affect human health.

### TPR-containing proteins in bacteria

Our analysis indicates that although TPR domains are present in bacterial virulence factors, the abundance of TPR-containing proteins in a proteome is not indicative of virulence. Rather, the specific function of an individual TPR-containing protein likely determines its capacity for virulence. This conclusion is consistent with the variable

nature of the amino acid sequence of the TPR repeats, which ultimately determines the interaction profile of the protein (*D'Andrea & Regan, 2003*). Further structural and molecular analysis of these domains within bacterial virulence factors has been suggested as a means to identify novel antibacterial drugs or the design of attenuated live vaccines that target the interaction between the host's cellular machinery and bacterial TPR domain (*Cerveny et al., 2013*).

## ARM-containing proteins in bacteria

In our analysis, the ARM superfamily is composed of 23 families of repeat domains (*SUPERFAMILY*). Many of these domains are composed of conserved structural features such that specific sequence and structural analysis is required to determine the exact identity of the repeat (*Andrade et al., 2001*). Within bacterial species, a HEAT-like domain, specifically the PBS lyase HEAT-like repeat, was the most common of the 23 families of ARM domains present in the ARM superfamily (Fig. S4). This domain is known to be present in phycocyanobilin lyases, which are a major component of phycobilisomes, a light harvesting complex in cyanobacteria (*Fairchild & Glazer, 1994*). However, this domain is also present in representative strains of all 25 bacterial classes analyzed (Table S6). PBS lyase HEAT-like domains are also present in many archaeal strains (7/8 analyzed, Table S7), and these domain-containing proteins function in chemotaxis as well as modify tRNA (*Miles et al., 2011*; *Schlesner et al., 2009*). It is important to note that the identification of ARM domains by prediction algorithms has been difficult and often misleading due to sequence variability within repeats, and that individual repeat analysis is important (*Kippert & Gerloff, 2004*; *Kippert & Gerloff, 2009*).

## Repeat domains in archaea

Similar to bacteria, a majority of archaea taxa contained tandem-repeat proteins and have, on average, a greater abundance of TPR-containing proteins than ARM and ANK-containing proteins (Figs. 1 and 2). However, archaea have a greater abundance of ARM-containing proteins, on average, than do bacterial species (Figs. 2B and 2C). The only class that includes species with no ARM or TPR-containing proteins is the Thermoprotei. While we are unsure of the proximate cause, this anomaly could be due to a phylogenetic contingency as Thermoprotei is the single class in our data set that is part of the phylum Crenarchaeota, rather than Euryarchaeota.

## Evolution of tandem repeats

The precise number of repeats in a repeat domain and their amino acid sequences are often variable between orthologs, although specific residues critical for the structure of the motifs remain conserved (*Javadi & Itzhaki, 2013*). The variation in repeat number is presumed to occur through tandem repeat domain duplication via recombination events resulting in repeat expansion (*Andrade et al., 2001*; *Marcotte et al., 1999*). A study on the evolution of the ANK-containing genes in *Wolbachia* endosymbionts of *Drosophila* indicated that both homologous and illegitimate recombination, as well as genomic flux (i.e., bacteriophage and transposable elements) were responsible for generating sequence

variability between ANK-containing gene orthologs (*Siozios et al., 2013*). This variability has been utilized as a diagnostic tool for *Wolbachia* strain fingerprinting (*Riegler et al., 2012*). Some repeat domain families, including the ANK domain, have been found to duplicate consistent numbers of domains instead of duplicating one repeat at a time (*Björklund, Ekman & Elofsson, 2006*). Since repeat domains are defined by their collective structure rather than the exact amino acid sequence of each repeat, a substantial amount of variation, or degeneracy, within a single repeat after duplication can be maintained without threatening the integrity of the protein domain (*Andrade et al., 2001*).

Domain duplication via recombination is error prone and allows for more rapid evolution than non-repeat containing proteins, which may provide the organism with a relatively quick means of adapting to new environments (*Marcotte et al., 1999*). Eukaryotes that contain more repeat-domain containing proteins than either bacteria or archaea may benefit from the increased sequence diversity (*Marcotte et al., 1999*). For example, the rapid evolution of leucine-rich repeats (LRR), a repeat domain present in many plant resistance (R) genes, appears to drive adaptations in plant innate immune systems (*Ellis, Dodds & Pryor, 2000*; *Liu et al., 2007*). In contrast to this general theory, a recent analysis on the evolution of tandem-repeat proteins in humans found that none of the analyzed proteins underwent a recent duplication or deletion, and a majority of human tandem repeat proteins (61%) were strongly conserved among all mammals (*Schaper, Gascuel & Anisimova, 2014*). Additionally, the substitution rate of tandem repeats themselves was 2.3 times lower than the sequence surrounding the repeat, indicating that the exact sequence of a protein's tandem repeat, and its subsequent structure and function, was conserved over time (*Schaper, Gascuel & Anisimova, 2014*). Either taxa vary in their adaptive capacity via repeat domain changes, or changes in individual repeat domains of specific proteins are advantageous during bouts of adaptation.

Whether or not repeat domains in bacteria and archaea evolved through convergent evolution, by descent with modification from a common ancestor, or acquired through horizontal gene transfer (HGT) has yet to be resolved. An analysis of the functional classes of eukaryotic, bacterial and archaeal repeat containing proteins showed that the more ancient the protein class shared between the domains of life, the more likely it would lack repeat proteins, and protein classes that were unique to eukaryotes were the most likely to contain repeat domains (*Marcotte et al., 1999*). This observation suggests that the acquisition of these repeat domains was a relatively recent evolutionary event. However, due to the apparent abundance of TPR repeats in all domains of life, it is likely that TPR domains did not independently evolve through convergent evolution (*Andrade et al., 2001*; *Ponting et al., 1999*). It was also originally postulated that bacterial ANK-containing proteins were obtained through HGT events, although this does not appear to explain how archaea and non-host associated microbes obtained their ANK-containing proteins (*Al-Khodor et al., 2010*; *Bork, 1993*; *Jernigan & Bordenstein, 2014*). Taken together with our analysis here, it does not appear that there is a specific mechanism for the evolution of all repeat domains in bacteria and archaea.

# PeerJ

### Funding

This research was funded by NIH awards F32 GM 100778 and 5T32HD007043-34 to Kristin K. Jernigan, and R01GM085163 to Seth R. Bordenstein. The funders had no role in study design, data collection and analysis, decision to publish, or preparation of the manuscript.

### Grant Disclosures

The following grant information was disclosed by the authors:
NIH awards: F32 GM 100778, 5T32HD007043-34, R01GM085163.

### Competing Interests

The authors declare there are no competing interests.

### Author Contributions

- Kristin K. Jernigan conceived and designed the experiments, performed the experiments, analyzed the data, wrote the paper, prepared figures and/or tables, reviewed drafts of the paper.
- Seth R. Bordenstein conceived and designed the experiments, analyzed the data, wrote the paper, reviewed drafts of the paper.

### Supplemental Information

Supplemental information for this article can be found online at http://dx.doi.org/10.7717/peerj.732#supplemental-information.

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
