# Peer review of "Tandem-repeat protein domains across the tree of life"

_PeerJ, doi:10.7717/peerj.732_

## Round 0.1 · original submission · Major Revisions

The procedure the authors made is correct, it can be replicated and everything seems fine. However, I would like to see a test, some evidence, or confidence values on how the repeat domains are predicted. I understand the authors rely on the SUPERFAMILY database and therefore on their predictions. However, this is the base of this research and some support on it could reinforce the validity of the analysis.

Additionally, would the results change if redundancy is eliminated from the dataset? I'm concerned with the fact that there is too much redundancy in the species they analyze and their proteomes. Specially, because of the percentage thresholds they set. I would advice to keep an eye on this and test whether the results differ. The thresholds also change from data set to data set.

Reviewer 1 ·

Basic reporting

Tandem-repeat protein domains across the Tree of Life

This manuscript is a follow-up on a previously published paper by the same authors where they described ankyrin (ANK) repeat domains across the tree of life. They present here a similar set of results, with the same methodologies, but now assessing the abundance of armadillo (ARM) and tetratricopeptide (TRP) tandem-repeat domains. They report that all repeat domains are present in almost every organism in all domains of the tree of life, with differences in abundance, and that it is expected to see more TRP containing proteins in virulence factors of pathogen bacteria, but didn’t.

The overall impression I get is that this paper is rather similar to the Ankyrin paper. It may be acceptable after major revision

Experimental design

The procedure the authors made is correct, it can be replicated and everything seems fine. However, I would like to see a test, some evidence, or confidence values on how the repeat domains are predicted. I understand the authors rely on the SUPERFAMILY database and therefore on their predictions. However, this is the base of this research and some support on it could reinforce the validity of the analysis.

Additionally, would the results change if redundancy is eliminated from the dataset? I'm concerned with the fact that there is too much redundancy in the species they analyze and their proteomes. Specially, because of the percentage thresholds they set. I would advice to keep an eye on this and test whether the results differ. The thresholds also change from data set to data set.

Validity of the findings

As stated above, their validity could be enhanced if the original predictions used are supported better. A "quality control" is needed to assess if the predictions are correct.

Their statistical analyses seem fine and make sense. I believe that it is also important to report negative results.

Additional comments

Major points:
The abstract gives no relevant information on the importance of finding these motifs in this fashion. For neophytes, the abstract is boring and does not incite to read the article. It can even be a little confusing.

Material and methods:
Does the inclusion of redundant species and proteomes bias the results?

Lines from 130, why would the domains be correlated? I think this correlation has no biological meaning and should be taken out of the manuscript, or otherwise specified why does the authors think this correlation is important? It is obvious that the presence of TPR is going to correlate with everything else since this domain is widely distributed across the whole tree of life.

I certainly enjoyed the discussion on the Mollicutes, since these organisms tend to fall out of generalities. However, the speculation about how the tandem repeat domain-containing proteins is related to their cell wall seems unsustainable. I believe that speculations this size without any evidence can be harmful to the way science is done. I would strongly advice to remove lines 224-226 unless the idea can be correctly supported.

Minor points:

The order of the figures is confusing, figure 3 appears before figure 2 in the text and it is distracting. Changing the order of figure 2 to 3 would be better.

The percentages in line 116 and 117 are way off, for bacterial genera, 308/544=56.6% and the authors report 76.8%, for the archaeal genera, 6/68=8.82% and 79.4 is reported. The numbers in the parenthesis must be wrong because the percentages correspond to the next part of the sentence.

The paragraph at line 160 needs to be revised. It is confusing, and the idea is not, so it can be written nicer.

Line 205, the last word “species” does not have an accent

Line 216, Change word Interesting, to interestingly

·

Basic reporting

No comments

Experimental design

Although, interesting results are shown, as the phylogeny rather than lifestyle or pathogenicity, is correlated to repeat domain abundance; or that pathogenic bacteria were not enriched in TPR-containing proteins, unlike previous reports. It’s hard to identify the main research question of the work. It is descriptive work, which lacks explanations or hypothesis to explain the data and results. In my opinion, this is the most important deficiency of the work. Hence, the article should be published until it resolves this void.

Validity of the findings

No Comments

Additional comments

Although, interesting results are shown, as the phylogeny rather than lifestyle or pathogenicity, is correlated to repeat domain abundance; or that pathogenic bacteria were not enriched in TPR-containing proteins, unlike previous reports. It’s hard to identify the main research question of the work. It is descriptive work, which lacks explanations or hypothesis to explain the data and results. In my opinion, this is the most important deficiency of the work. Hence, the article should be published until it resolves this void.

---

## Round 0.2 · accepted · Accept

The authors did all the suggested corrections, I fell the manuscript is now well rounded.